# Differential Proteomic Analysis of Extracellular Vesicles Produced by *Granulicatella adiacens* in Biofilm vs. Planktonic Lifestyle

**DOI:** 10.3390/dj13120557

**Published:** 2025-11-26

**Authors:** Maribasappa Karched, Sarah Alkandari

**Affiliations:** 1Oral Microbiology Research Laboratory, Department of Bioclinical Sciences, College of Dentistry, Health Sciences Centre, Kuwait University, P.O. Box 24923, Safat 13110, Kuwait; 2Microbiome Laboratory, Guy’s Hospital, King’s College, London SE1 9RT, UK; sarah.alkandari@kcl.ac.uk

**Keywords:** *Granulicatella*, proteomics, biofilm, planktonic, oral infections

## Abstract

**Background:** Gram-positive bacteria, once considered incapable of producing extracellular vesicles (EVs) due to their thick peptidoglycan layer, are now known to secrete EVs that transport virulence factors and modulate host immunity. These EVs contribute to bacterial pathogenicity by facilitating biofilm formation, immune evasion, and inflammation. *Granulicatella adiacens*, an oral commensal associated with infective endocarditis, represents a clinically relevant model to study EV-mediated virulence. **Objectives:** This study’s aim was to investigate whether the proteomic composition and immunomodulatory activity of *G. adiacens* EVs differ between biofilm and planktonic lifestyles, thereby contributing to distinct pathogenic behaviours. **Methods:** EVs isolated from *G. adiacens* CCUG 27809 cultures were characterized using nano LC-ESI-MS/MS, followed by comprehensive bioinformatic and cytokine assays. **Results:** Quantitative proteomic profiling identified 1017 proteins, revealing distinct signatures between biofilm- and planktonic-derived EVs. Principal component analysis showed clear segregation between the two states, with biofilm EVs enriched in proteins linked to stress adaptation, adhesion, and structural integrity, while planktonic EVs exhibited growth- and metabolism-related proteins. A total of 114 virulence-associated proteins were identified, including several novel candidates. Functionally, EVs from both conditions significantly induced pro-inflammatory cytokines IL-8 and IL-1β in a dose-dependent manner (*p* < 0.05), whereas IL-17 remained unchanged. **Conclusions:** *G. adiacens* EVs exhibit lifestyle-dependent proteomic and immunomodulatory differences, underscoring their role in host–pathogen interactions and endocardial infection. These findings provide a foundation for future mechanistic and in vivo studies exploring EV-mediated virulence and potential therapeutic modulation.

## 1. Introduction

Transient bacteremia following dental procedures can allow oral microorganisms to enter the bloodstream, occasionally leading to infective endocarditis [1,2]. Among these, *Granulicatella* species—facultatively anaerobic, Gram-positive cocci—have emerged as opportunistic pathogens associated with oral and systemic infections, notably infective endocarditis [3,4,5]. Although *G. adiacens* is part of the normal oral microbiota [6], it has been implicated in dental caries, endodontic infections, and periodontitis [7,8,9], suggesting that under specific conditions it can shift from commensalism to pathogenicity.

Bacterial extracellular vesicles (EVs) have emerged as crucial mediators of intercellular communication and virulence. Gram-negative bacteria are well known to produce outer membrane vesicles (OMVs) derived from membrane blebbing [10,11], but recent studies confirm that Gram-positive species also release EVs despite their thick peptidoglycan walls [12,13,14] and recent reviews have updated mechanisms of biogenesis, composition, and roles of Gram-positive EVs in pathogenesis and host modulation [15,16]. These Gram-positive EVs can carry a range of components—including enzymes, toxins, lipids, and nucleic acids—and have been implicated in modulating host immune responses, promoting biofilm formation, and enhancing bacterial survival under stress [14,17]. Furthermore, external interventions such as antimicrobial agents [18,19,20] and photobiomodulation (e.g., low-level laser therapy) [21], can alter oral microbiota composition and host responses, potentially influencing EV production and signalling, underscoring their clinical significance.

In line with proteomic differences between biofilm and planktonic cells [22,23], proteomic analyses comparing EVs from biofilm and planktonic conditions have highlighted quantitative differences in composition that likely reflect the physiological conditions [24]. In the context of oral cavity, previous studies have profiled the secretome and EV proteome of *G. adiacens* [25,26], yet the impact of bacterial lifestyle—biofilm versus planktonic growth—on EV composition and virulence potential remains unexplored. Biofilm formation is known to enhance microbial resistance and virulence [27,28,29]; therefore, EVs released under these conditions may differ functionally and structurally from those produced in planktonic cultures.

This study aims to test the hypothesis that EVs released by *G. adiacens* during biofilm growth differ proteomically and functionally from those produced in planktonic conditions, contributing to distinct mechanisms of virulence and host immune modulation. To address this, we isolated and characterized EVs from both lifestyles using high-resolution quantitative proteomics and evaluated their cytokine-inducing capacity in vitro. This comparative approach provides the first insight into lifestyle-specific EV-mediated pathogenic pathways in *G. adiacens*.

## 2. Materials and Methods

### 2.1. Bacterial Strain and Culture Conditions

The reference strain *Granulicatella adiacens* CCUG 27809 was used for all experiments. This well-characterized strain was grown on chocolate blood agar (CBA) supplemented with 0.001% pyridoxal hydrochloride and incubated at 37 °C under 5% CO_2_ for 48 h. A loopful of colonies was subsequently inoculated into brucella broth containing 0.001% pyridoxal hydrochloride and incubated under identical conditions for another 48 h. Three independent biological replicates were prepared for all experiments.

### 2.2. Biofilm and Planktonic Cultures

Biofilm and planktonic cultures were prepared following a previously described method with minor modifications [30]. Bacterial cells from agar plates were suspended in brucella broth and washed once (5000× *g*, 5 min) to ensure purity. The cell pellet was resuspended in 1 mL of fresh broth, and the optical density (OD_600_) was adjusted to 1.

For biofilm formation, 100 µL of this standardized suspension was inoculated into each well of a 24-well plate containing 900 µL of brucella broth. Planktonic cultures were initiated by adding 100 µL of the same suspension to 900 µL of broth in sterile microfuge tubes. Wells and tubes containing only brucella broth served as negative controls. Cultures were incubated at 37 °C under 5% CO_2_ for 24 h.

### 2.3. Isolation of Extracellular Vesicles (EVs)

EVs were isolated using a differential centrifugation and filtration protocol [25] with modifications. Broth cultures were centrifuged at 5000× *g* for 10 min to remove cells (Eppendorf, Hamburg, Germany). The supernatant was filtered through a 0.22 µm syringe filter (Millipore, Darmstadt, Germany) to eliminate residual bacteria and ultracentrifuged at 125,000× *g* for 3 h at 4 °C (Beckman, Indianapolis, IN, USA). The resulting pellet was resuspended in sterile phosphate-buffered saline (PBS) and stored at −20 °C until use.

### 2.4. Scanning Electron Microscopy (SEM)

EV samples were fixed in PBS containing 3% glutaraldehyde for 2 h at room temperature, followed by overnight storage at 4 °C. Post-fixation was carried out with 1% osmium tetroxide for 2 h. Samples were dehydrated through a graded acetone series (30–100%), critical-point dried, mounted on aluminum stubs with carbon tape, sputter-coated with gold, and stored in a desiccator. Imaging was performed using a Zeiss Leo Supra 50 VP field emission SEM (Carl Zeiss, Oberkochen, Germany).

### 2.5. Protein Quantification and SDS-PAGE

Protein concentrations were determined using the Quick Start™ Bradford Microplate Assay (Bio-Rad, Hercules, CA, USA). Equal protein quantities were denatured in 2× Laemmli buffer (100 °C, 5 min), centrifuged (5000× *g*, 5 min), and separated by SDS-PAGE (12% gels, 120 V). Silver staining was used for visualization, and images were captured using a ChemiDoc™ MP Imaging System (Bio-Rad, USA).

### 2.6. Sample Preparation for Mass Spectrometry

A total of 25 µg of each EV protein sample was reduced with 5 mM TCEP, alkylated with 50 mM iodoacetamide, and digested overnight with trypsin (1:50 enzyme–substrate, 37 °C). Peptides were purified using C18 silica cartridges, dried under vacuum, and resuspended in buffer A (2% acetonitrile, 0.1% formic acid).

### 2.7. LC-MS/MS Analysis

Peptides were analyzed on an Easy-nLC 1000 system coupled to an Orbitrap Exploris 240 mass spectrometer (Thermo Fisher Scientific, Waltham, MA, USA). Peptides (1 µg) were loaded onto a Picofrit column (1.8 µm resin, 15 cm) and eluted with a 0–38% gradient of buffer B (80% acetonitrile, 0.1% formic acid) at 500 nL/min for 96 min. The instrument was operated in data-dependent acquisition mode (top 20) with a dynamic exclusion of 30 s. MS spectra were acquired at 60 K resolution (*m*/*z* 375–1500), and MS2 spectra at 15 K resolution.

Raw data have been deposited in the PRIDE repository (Project ID: PXD059541; username: reviewer_pxd059541@ebi.ac.uk; password: gyAZOgM9BoHt).

### 2.8. Data Processing

Raw files were analyzed using Proteome Discoverer v2.5 against the *G. adiacens* UniProt database. Search parameters included a precursor mass tolerance of 10 ppm and fragment tolerance of 0.02 Da. Trypsin/P was specified as the digestion enzyme, with carbamidomethyl (C) as a fixed modification and oxidation (M) and N-terminal acetylation as variable modifications. Peptide-spectrum matches and protein-level false discovery rates (FDR) were controlled at 1%.

### 2.9. Data Pre-Processing and Quality Control

Proteins were retained only if quantified in ≥2 of 3 biological replicates under at least one condition (biofilm or planktonic). Abundance values were log_2_-transformed and missing data imputed using the “MinProb” method, assuming left-censored missingness. This approach models low-abundance proteins while minimizing variance inflation.

### 2.10. Differential Protein Expression Analysis

Differentially expressed proteins between biofilm and planktonic EVs were identified using the *limma* package (R/Bioconductor), which applies empirical Bayes moderation to improve statistical power. Adjusted *p*-values (Benjamini–Hochberg correction) < 0.05 and |log_2_ fold change| > 1.5 were considered significant.

### 2.11. Data Visualization and Functional Analysis

Principal component analysis (PCA), volcano plots, and hierarchical clustering were performed using R (v4.2) packages FactoMineR, EnhancedVolcano, and ComplexHeatmap, respectively. Gene Ontology (GO) enrichment was carried out in ExpressAnalyst using hypergeometric testing with FDR correction (adjusted *p* < 0.05). Protein–protein interaction networks were generated using STRING v12.5 (confidence > 0.7) to identify functional modules and hub proteins.

### 2.12. Cytokine Quantification

Peripheral blood mononuclear cells (PBMCs) were stimulated with EVs, and cytokines (IL-8, IL-1β, IL-17) were quantified using Quantikine^®^ ELISA kits (R&D Systems, Minneapolis, MN, USA) following the manufacturer’s instructions. Absorbance was measured at 450 nm with wavelength correction at 570 nm (iMark™ Microplate Reader, Bio-Rad). All assays were performed in duplicate, and cytokine concentrations were calculated from standard curves.

### 2.13. Statistical Analysis

For proteomics data, differential abundance testing was performed as described above. Cytokine data, which were non-normally distributed, were compared using the nonparametric Mann–Whitney U test (*p* < 0.05). Statistical analyses were performed using SPSS v25.0.

### 2.14. Ethical Approval

This study was approved by the ethical committee of the Health Sciences Center, Kuwait University (DR/EC/3413, Date: 31 October 2018), and has been carried out in full accordance with the World Medical Association Declaration of Helsinki. The blood donor received written information about the nature and purposes of the study and a written informed consent was obtained upon his/her approval to participate.

## 3. Results

### 3.1. EV Preparation from G. adiacens Biofilm and Planktonic Cells

Following previously established protocols in our laboratory, vesicles were isolated from bacterial cultures of the reference strain *G. adiacens* CCUG 27809. Determination of the protein concentration in EV preparations revealed values ranging from 400 to 600 µg across three independent isolations. SDS–PAGE analysis (Figure 1A) showed distinct banding patterns between vesicles derived from biofilm and planktonic growth. In silico analysis of the protein sequences using a 2D-gel analysis tool indicated that most cytoplasmic proteins had isoelectric points (pI) between 4.5 and 6.0, whereas most secreted proteins were in the pI range of 8.0 to 9.6 (Figure 1B). Scanning electron microscopy (SEM) of EV preparations showed no observable morphological differences between the two sample types in terms of size or abundance (Figure 1C).

### 3.2. Differential Protein Expression in Vesicles Produced Under Biofilm and Planktonic Conditions

To define molecular adaptations of *G. adiacens* to different growth modes, quantitative proteomic profiling was performed on extracellular vesicles (EVs) isolated from biofilm and planktonic cultures. The volcano plot (Figure 2A) highlighted proteins significantly upregulated in biofilm (blue) and planktonic (red) conditions, while grey points represented non-significant changes. Principal component analysis (PCA) revealed strong separation between the two states, with the first component explaining ~75% of total variance. Pearson’s correlation heatmaps (Figure 2B) demonstrated large co-regulated protein modules, indicating that the biofilm–planktonic transition involves coordinated regulation of functional networks rather than random protein-level changes.

Mass spectrometry identified 1017 proteins in biofilm-derived EVs and 1048 in planktonic EVs. Consistency across biological replicates was confirmed by line plots showing log_2_-transformed intensity profiles for significantly upregulated and downregulated (Appendix A) proteins, both displaying uniform expression trends. Proteomic analysis revealed distinct functional profiles between biofilm and planktonic vesicles. Biofilm-associated EVs were enriched in proteins related to cell wall and peptidoglycan synthesis (e.g., Lipid II isoglutaminyl synthase subunits GatD and MurT, peptidoglycan glycosyltransferase, LPXTG-motif cell wall anchor proteins), ATP synthase components, and energy metabolism enzymes. These trends suggest an active structural and metabolic adaptation supporting matrix formation and sessile survival. In contrast, planktonic vesicles showed upregulation of proteins involved in translation, redox metabolism, and cell division (e.g., ribosomal proteins, FtsZ, FtsA, redoxin family enzymes), reflecting rapid growth and motility.

Gene Set Enrichment Analysis (GSEA) further supported these distinctions (Appendix A). Pathways related to cell wall metabolism (Enrichment Score = 0.106) and stress response (Enrichment Score = 0.083) were enriched in biofilm EVs, whereas transport systems (Enrichment Score = –0.373) and transcription machinery (Enrichment Score = –0.517) were enriched in planktonic EVs.

### 3.3. Differentially Expressed Proteins Exhibit Consistent and Robust Expression Patterns

The robustness of the proteomic signatures was confirmed through several quality-control analyses. Unsupervised hierarchical clustering of Z-score normalized abundances of significantly differentially expressed proteins segregated samples strictly according to their growth condition (Figure 3). All biofilm replicates clustered together, distinct from planktonic replicates, demonstrating systematic and reproducible proteomic divergence.

Row-wise clustering revealed two major co-regulated protein blocks: one consistently upregulated in biofilms (red) and the other downregulated (blue). Sample-to-sample correlation analysis (Appendix A) showed strong within-group correlations (r ≈ 1.0) and low between-group correlations, confirming excellent reproducibility. Violin plots of log_2_-transformed protein abundances showed consistent distributions across all samples, ruling out normalization bias (Appendix A).

Representative scatter plots (Appendix A) for select proteins showed statistically significant differences (*p* < 0.01) between biofilm (blue) and planktonic (orange) replicates, validating the robustness of the differential expression patterns.

### 3.4. Protein Overlaps and Distribution Between Vesicles from Biofilm and Planktonic Conditions

Protein overlap analysis (Figure 4) highlighted substantial similarity between the two proteomes: 96% (979 proteins) were shared between biofilm and planktonic EVs, while 22 and 17 proteins were unique to biofilm and planktonic conditions, respectively (Figure 4A). This suggests that transitions between growth states primarily reflect quantitative modulation of a shared proteome rather than qualitative replacement.

The biofilm-unique proteins included magnesium transporters (MgtE), ABC transporters, cyclic-di-AMP phosphodiesterase, and metallo-β-lactamase—proteins likely contributing to biofilm stability and stress resilience. Planktonic-specific proteins, such as the 60-kDa chaperonin, alcohol dehydrogenase, and SepF, were associated with growth and division. These findings identify biofilm-specific molecular targets (e.g., MgtE, ABC transporters) that may inform anti-biofilm strategies.

### 3.5. Gene Ontology and Functional Annotation of Differentially Expressed Proteins

Network analysis (Figure 5A) revealed a central “Metabolic Process” hub linking major branches such as proteolysis, carbohydrate derivative metabolism, and organic acid metabolism—indicating coordinated regulation of energy and matrix-related processes. Biofilm vesicles emphasized protein degradation and cell envelope processes, while planktonic vesicles favoured biosynthesis and cellular development.

Heatmaps (Figure 5B) showed distinct enrichment patterns: biofilms exhibited elevated activity in proteolysis and membrane-associated processes, whereas planktonic cells demonstrated higher activity in metabolic and growth-related functions.

### 3.6. Proteins with Predicted Virulence Potential

Table 1 lists 114 proteins with predicted virulence potential, several of which are well-documented in bacterial pathogenicity. Of the 22 biofilm-unique proteins, six were predicted to be virulent, compared to none in the planktonic-unique set. These included Type VII secretion system components (e.g., YukD, EsaA), LPXTG-motif anchor proteins, metallo-β-lactamase, and cell wall-binding repeat proteins. High-scoring ABC transporters and several uncharacterized proteins were also identified, suggesting potential new virulence factors warranting further investigation. Overall, biofilm EVs were enriched in surface-associated and secretion-related virulence determinants.

### 3.7. Proinflammatory Potential of EV Preparations

The proinflammatory effects of the EV preparations were evaluated using quantitative ELISA kits (R&D Systems). As shown in Figure 6, significant differences (*p* < 0.05) were observed in IL-8 and IL-1β levels between biofilm- and planktonic-derived EVs, with a dose-dependent increase in both cytokines. IL-17 levels did not differ significantly between the two conditions.

## 4. Discussion

The present study provides the first comprehensive characterization of extracellular vesicles (EVs) produced by *G. adiacens* under biofilm and planktonic growth conditions, revealing distinct proteomic and immunostimulatory profiles associated with each lifestyle. These findings support the growing evidence that bacterial EVs are key mediators of virulence, intercellular communication, and environmental adaptation [11,12,14] and recent studies further highlight their diverse immunomodulatory roles and clinical potential in Gram-positive infections [31,32]. While earlier work examined protein secretion in *Granulicatella* species [25], this is the first report describing proteomic differences in EVs derived from biofilm versus planktonic *G. adiacens*, offering novel insights into its pathogenic potential.

SDS-PAGE analysis revealed subtle but reproducible differences in EV protein profiles between biofilm and planktonic cultures, suggesting condition-specific variation in protein cargo. In silico 2D-gel mapping indicated that cytoplasmic proteins clustered predominantly in the acidic pI range (4.5–6.0), whereas secreted proteins were enriched in the alkaline range (8.0–9.6), consistent with trends observed in other Gram-positive bacteria. SEM imaging showed no morphological differences between biofilm- and planktonic-derived vesicles, implying that their functional divergence arises primarily from molecular composition rather than structural variation.

Proteomic analysis identified 1017 proteins, with 96% shared between both conditions. Nevertheless, differential abundance patterns and a subset of condition-specific proteins (22 biofilm-unique, 17 planktonic-unique) suggest specialized adaptations. This near-complete overlap implies that the shift between these lifestyles relies primarily on quantitative changes (up- or down-regulation) in a shared proteome rather than a largescale protein replacement. However, unique proteins in each state, though small in number—could play pivotal roles in biofilm formation or planktonic lifestyle, acting as specialized molecular switches for these distinct modes of existence. Biofilm-specific proteins such as magnesium transporter (mgtE) and Cyclic-di-AMP phosphodiesterase have been shown to play important roles in the regulation of type III secretion system in different bacteria [33,34]. Further, deletion of mutL, a protein involved in DNA mutation repair was found to affect adhesion and biofilm formation leading to an overall reduced virulence [35]. Proteins with metallo-beta lactamase domain can confer resistance to a broad range of beta-lactam antibiotics. Its presence in biofilm vesicles indicates a mechanism for antibiotic inactivation [36].

Biofilm EVs were enriched in proteins linked to cell-wall biogenesis and metabolic persistence, such as GatD, MurT, and LPXTG-anchored surface proteins. The prominence of MurT/GatD pathway components—known to be essential for cell-wall amidation and antibiotic tolerance in other pathogens including *S. aureus* and *M. tuberculosis* [37,38,39]—suggests a similar role in maintaining biofilm integrity and survival. Elevated levels of stress-response chaperones (DnaK, trigger factor) and ribosome hibernation factors further align with the biofilm’s demand for long-term stability under nutrient-limited conditions.

Conversely, planktonic EVs showed enrichment in ribosomal proteins, FtsZ, and transcriptional regulators (SigA, Rho), consistent with rapid proliferation and metabolic flexibility. The presence of redox-active enzymes such as glycerol dehydrogenase implies enhanced energy metabolism to sustain a free-living state. PCA and hierarchical clustering analyses further confirmed clear proteomic segregation between biofilm and planktonic EVs, reflecting extensive physiological reprogramming during the biofilm–planktonic transition.

Gene Ontology (GO) enrichment analyses revealed that biofilm EVs are functionally biased toward proteolysis and carbohydrate metabolism, likely supporting matrix remodelling and energy production, whereas planktonic EVs favoured metabolic versatility and cell development pathways. Notably, 114 virulence-associated proteins—including Type VII secretion components (YukD, EsaA), cell-wall hydrolases, and antibiotic resistance determinants (metallo-β-lactamase)—were identified, highlighting the potential contribution of *G. adiacens* EVs to host–pathogen interactions and disease progression.

Functionally, EVs from both growth states triggered significant induction of IL-8 and IL-1β in human PBMCs, with biofilm-derived EVs eliciting a stronger proinflammatory response. This selective cytokine activation is consistent with previous observations that biofilm bacteria provoke heightened innate immune signalling [12,40]. Interestingly, IL-17 levels remained unchanged, suggesting a targeted immune modulation rather than broad activation. IL-17 is an important cytokine in the mucosal immunity. The observed similarity in IL-17 stimulation by biofilm and planktonic EVs is likely because they share similar outer envelope components, such as bacterial cell wall elements and surface proteins, which are known to induce this key mucosal immunity cytokine [41,42]. The observed cytokine patterns are relevant to *G. adiacens* pathogenesis, given the roles of IL-8 and IL-1β in infective endocarditis and oral infections [43,44]. While TNF-α and IL-6 were not assessed in this study, future cytokine panels may include these key inflammatory mediators to better define the immunomodulatory spectrum of *G. adiacens* EVs.

Extensive research on bacterial membrane vesicles has established that a diverse array of well-characterized toxins and non-toxin virulence factors are secreted via vesicle pathway [45,46,47]. In contrast to their soluble counterparts, vesicle-associated virulence factors are uniquely shielded from degradation by host proteases [11]. Furthermore, vesicles facilitate the targeted delivery of these factors as concentrated packages to host cells and tissues, thereby amplifying damage at specific sites. Similar to oral bacteria implicated in infective endocarditis—such as *Aggregatibacter actinomycetemcomitans* [48], *Kingella kingae* [49], and others [25,50,51]—it is plausible that *G. adiacens* similarly utilizes its extracellular vesicles, which are enriched with numerous putative virulence proteins, in the pathogenesis of this infection.

## 5. Limitations and Future Directions

This study was conducted using EVs derived from the *G. adiacens* reference strain CCUG 27809 to ensure experimental reproducibility. The other limitation is that functional control experiments using heat- or protease-treated EVs were not performed in this study and will be addressed in future work to confirm that cytokine induction is protein-mediated. Future investigations should validate these findings using multiple clinical isolates to assess strain-specific variability. Additionally, while the proteomic and immunostimulatory characterizations provide a strong foundation, further in-depth analyses—including mechanistic studies with modified or digested EVs and in vivo infection models—are warranted to establish causal relationships between EV composition and virulence. Further research exploring the molecular mechanisms through which EVs mediate host–pathogen communication could also elucidate novel therapeutic targets.

## 6. Conclusions

This work demonstrates that *G. adiacens* produces extracellular vesicles with distinct molecular and immunological characteristics depending on its growth mode. Biofilm-derived EVs exhibit enrichment in structural and stress-response proteins, whereas planktonic EVs reflect a metabolically active state. The presence of multiple virulence-associated proteins and proinflammatory activity underscores the potential contribution of EVs to *G. adiacens* pathogenicity. This is consistent with recent reviews that discuss EV-mediated modulation of antibiotic tolerance and the potential diagnostic/therapeutic implications of bacterial EVs in clinical infections [15,32]. By elucidating these vesicular adaptations, our study provides a foundation for future mechanistic and translational research aimed at understanding and mitigating infections caused by this understudied organism.

## Figures and Tables

**Figure 1 dentistry-13-00557-f001:**
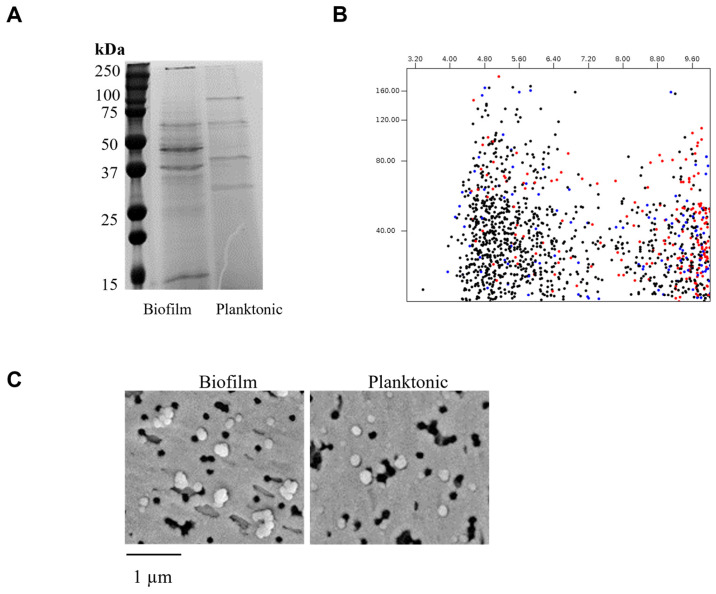
Analysis of *G. adiacens* extracellular vesicles. Vesicle preparations containing standard amounts of protein were run on an SDS-PAGE gel electrophoresis (Panel (**A**)) and in silico 2D gel visualization (Panel (**B**)). Scanning electron microscopy was performed for the vesicle preparations (Panel (**C**)).

**Figure 2 dentistry-13-00557-f002:**
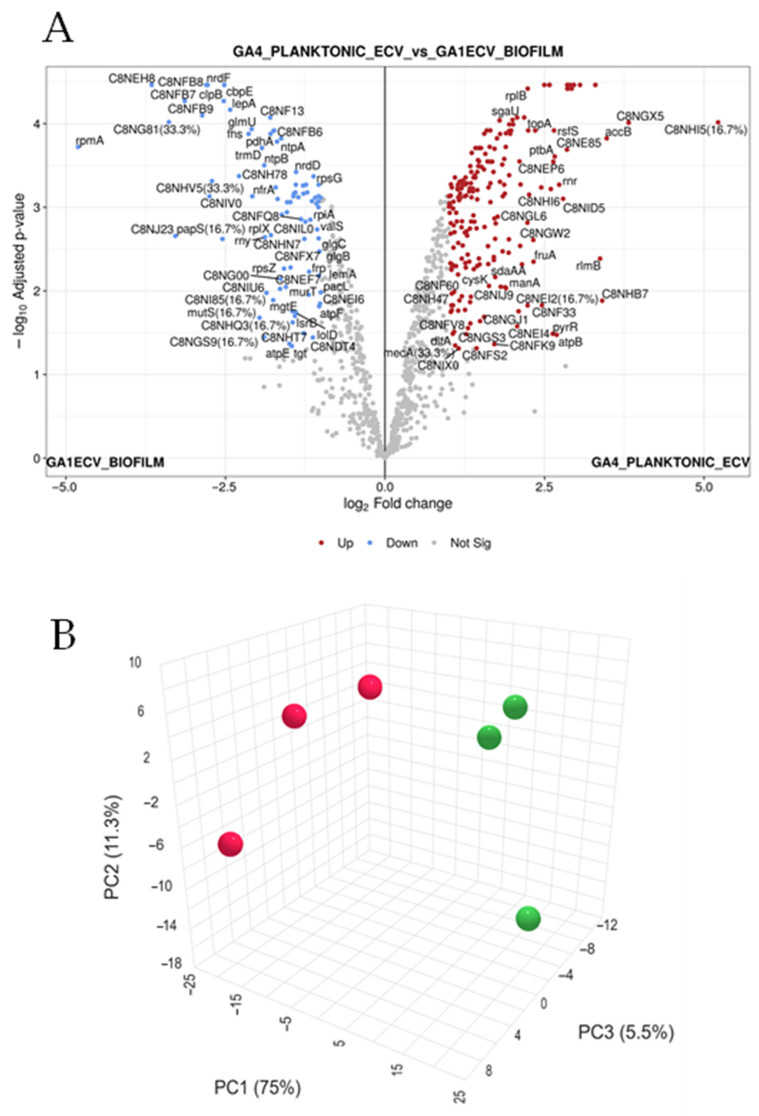
Global proteomic landscape of *G. adiacens* EVs from biofilm and planktonic cultures. This figure presents a global differential proteomic analysis, highlighting the distinct molecular signatures of extracellular vesicles (EVs) from *G. adiacens* grown in biofilm versus planktonic states. (**A**) Volcano plot illustrating the magnitude of protein expression change (*x*-axis: log2-FC) versus statistical significance (*y*-axis: −log10 adjusted *p*-value). Proteins significantly upregulated in planktonic EVs are shown as red points, those upregulated in biofilm EVs are blue points, and non-significantly changed proteins are grey points. Significance was determined by an adjusted *p*-value < 0.05 and an absolute log2-FC > 1.5. Key differentially expressed proteins are labelled. (**B**) Three-dimensional Principal Component Analysis (PCA) plot visualizing the global proteomic variance between samples. The plot demonstrates clear spatial separation of the biofilm (red spheres) and planktonic (green spheres) sample clusters. The first three principal components (PC1, PC2, and PC3) account for 75%, 11.3%, and 5.5% of the total variance, respectively, indicating distinct proteomic profiles. Statistical analysis was performed using a linear model with Benjamini–Hochberg correction for multiple testing. Red dots indicate biofilms while the green ones indicate planktonic samples.

**Figure 3 dentistry-13-00557-f003:**
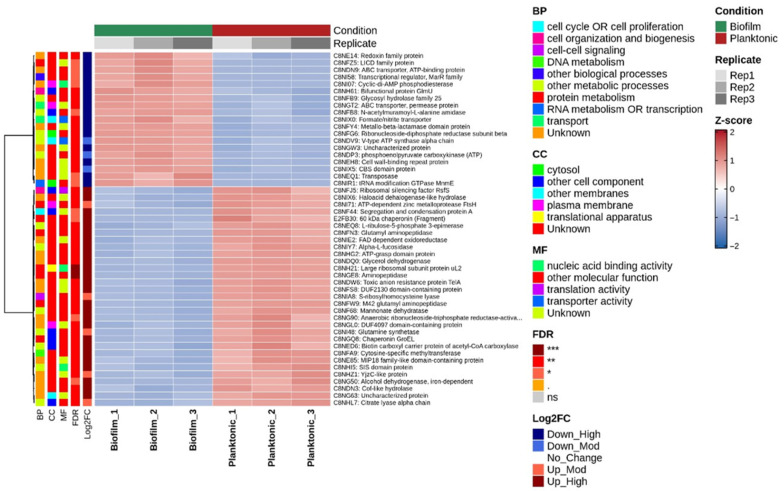
Hierarchical clustering of differentially expressed EV proteins. This figure provides a detailed examination of the expression patterns of significantly altered proteins, confirming the robustness and reproducibility of the proteomic data. A, Heatmap displaying the Z-score normalized, log_2_-transformed abundance of significantly differentially expressed proteins. Rows represent individual proteins and columns represent biological replicates (*n* = 3 per condition). The color key indicates relative abundance (blue: low abundance; red: high abundance). Unsupervised hierarchical clustering (Ward.D2 method, Euclidean distance) was applied to both samples (columns) and proteins (rows). The column dendrogram demonstrates a clear distinction between biofilm and planktonic samples, signifying their disparate EV proteomes. Annotations for Gene Ontology (GO) biological process, false discovery rate (FDR), and log_2_FC are provided. *: FDR < 0.05, **: FDR < 0.01, ***: FDR < 0.001, n.s.: not significant.

**Figure 4 dentistry-13-00557-f004:**
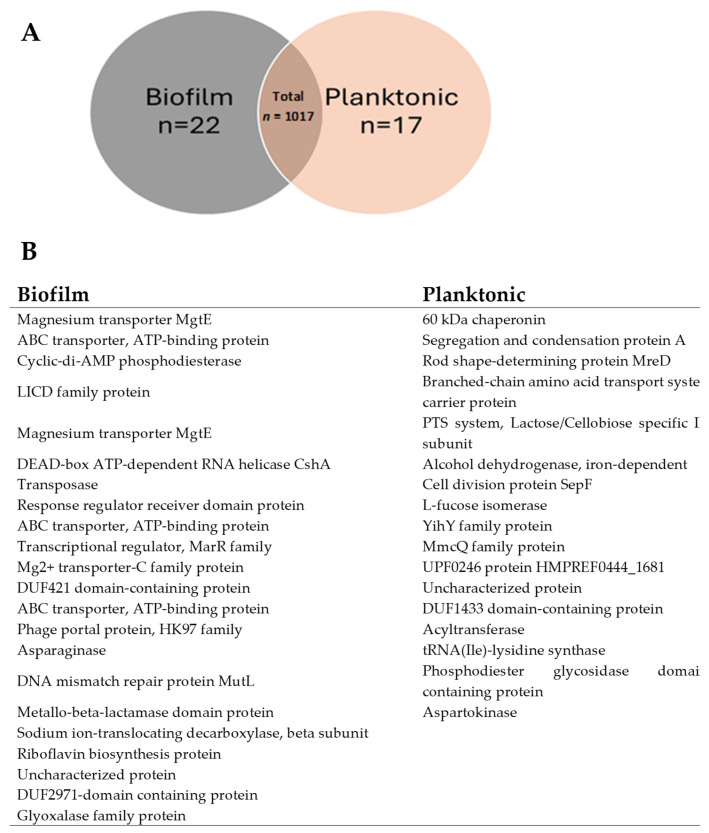
Protein Overlap and Distribution Between BIOFILM and PLANKTONIC States in *G. adiacens.* This figure dissects the composition of the biofilm and planktonic EV proteomes, highlighting both shared and unique protein components. (**A**) Venn diagram illustrating the overlap in protein expression between biofilm (GA1ECV_biofilm, red) and planktonic (GA4_planktonic_ECV, blue) conditions. Of the total 1017 proteins, 979 (96%) are shared between the two conditions, while 22 (2.16%) are unique to biofilm and 17 (1.67%) to planktonic. The count of proteins in each category is color-coded to indicate protein abundance, with higher counts shown in darker blue. (**B**) Table listing a selection of functionally relevant proteins unique to either the biofilm or planktonic state. Biofilm-unique proteins include factors associated with environmental adaptation and stress, such as the magnesium transporter MgtE and the DNA mismatch repair protein MutL. Planktonic-unique proteins include factors related to active proliferation, such as the cell division protein SepF and a 60 kDa chaperonin. These unique proteins may function as specialized molecular determinants for each distinct mode of existence. The overlap between biofilm and planktonic states was determined using a Venn diagram, and the protein counts were visualized in a bar plot to show the number of shared and unique proteins between conditions.

**Figure 5 dentistry-13-00557-f005:**
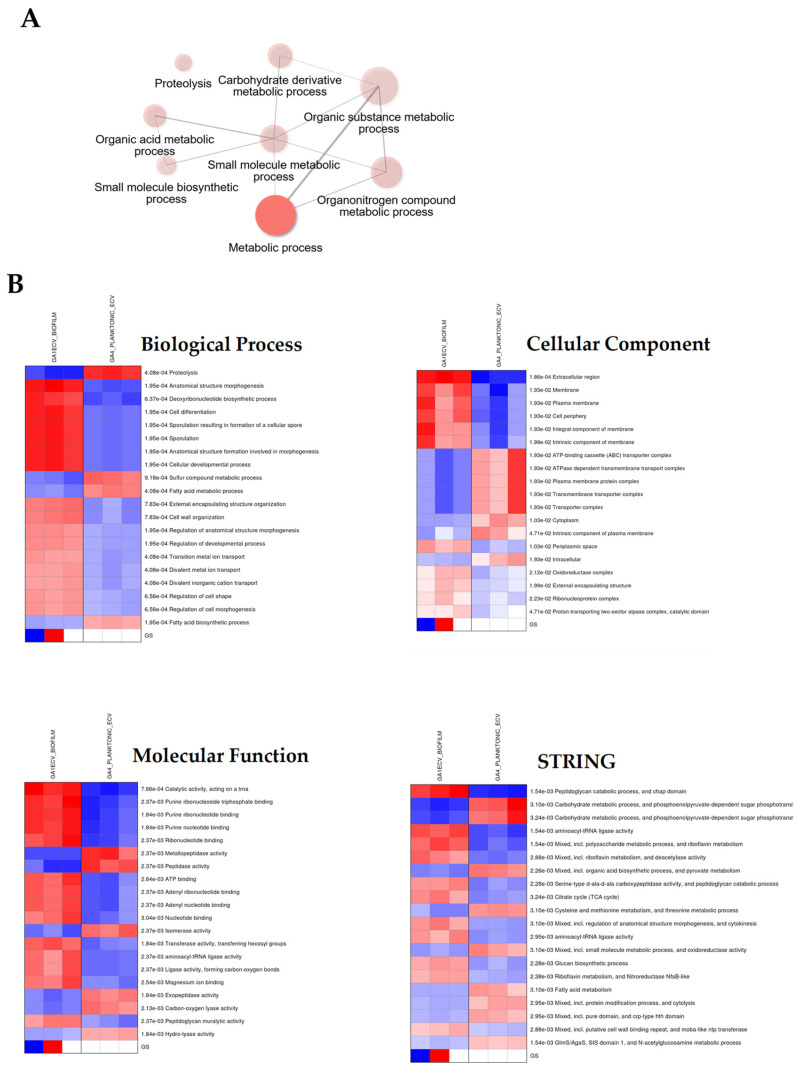
Gene Ontology and Functional Annotation of Differentially Expressed Proteins in *G. adiacens.* Panel (**A**): Network diagram illustrating the enrichment of gene ontology (GO) biological processes (BP) related to metabolic process. Proteins that are differentially expressed between biofilm (GA1ECV_biofilm, red) and planktonic (GA4_planktonic_ECV, blue) conditions are linked to specific biological processes. The sizes of the nodes correspond to the number of proteins associated with each process, with the largest node representing metabolic process. other enriched terms include proteolysis, small molecule metabolic process, and carbohydrate derivative metabolic process. Panel (**B**): Heatmaps of the top enriched GO terms for biofilm and planktonic conditions across three categories: The color intensity corresponds to the level of enrichment, with darker red and blue indicating more significant terms. Gene Ontology enrichment analysis was performed to identify biological processes, cellular components, and molecular functions enriched in differentially expressed proteins between the biofilm and planktonic conditions. A *p*-value threshold of 0.05 was used for statistical significance, and the results were visualized using network diagrams and heatmaps. Proteins were clustered based on their GO term associations, and hierarchical clustering was applied to group related terms.

**Figure 6 dentistry-13-00557-f006:**
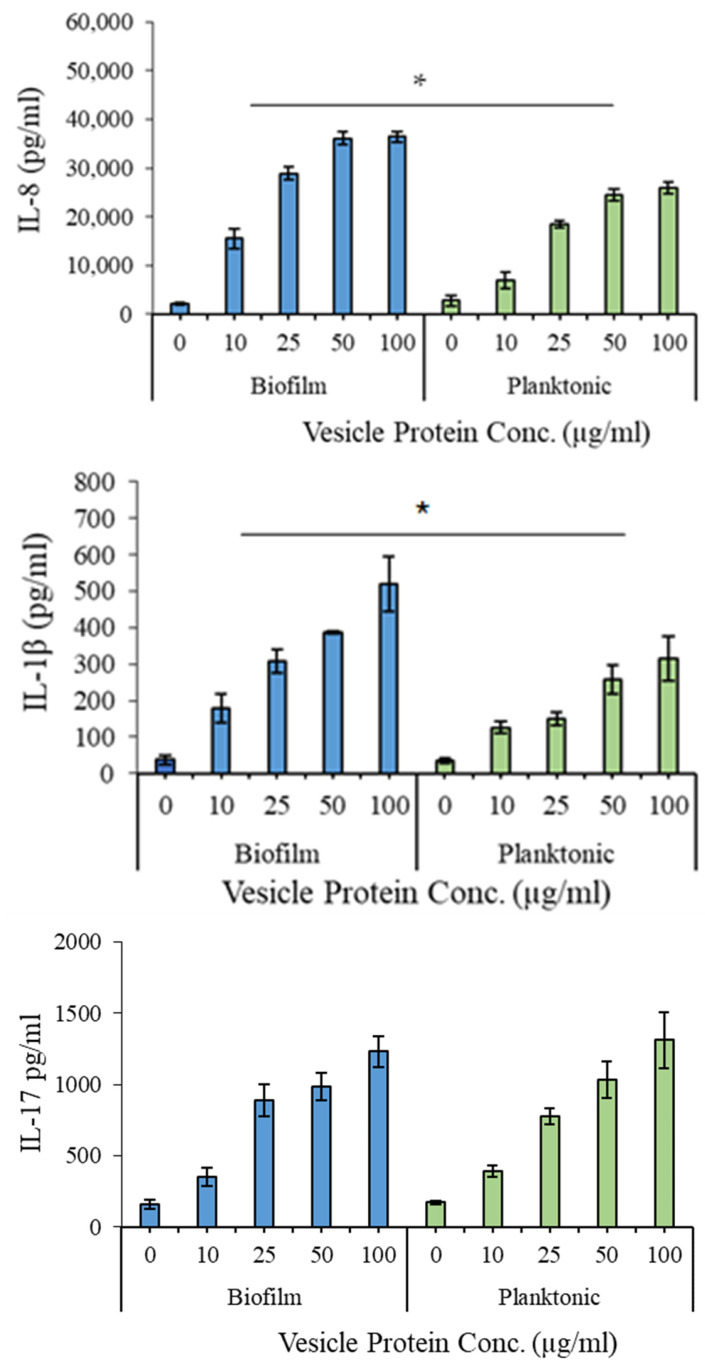
Proinflammatory potential of the extracellular vesicles produced by *G. adiacens* in biofilm and planktonic growth. Vesicle preparations at the concentrations 10, 25, 50, and 100 µg/mL were used for stimulating human PBMCs. Cytokine induction was considered significantly different at * *p* < 0.05.

**Table 1 dentistry-13-00557-t001:** **Predicted virulent proteins.**

S. No.	Protein Name	Predicted Scores
1	DUF2130 domain-containing protein	0.9537
2	Glycosyl hydrolase family 25	1.0436
3	Aminopeptidase	0.632
4	Toxic anion resistance protein TelA	0.9069
5	SIS domain protein	1.0157
6	Large ribosomal subunit protein uL2	0.5151
7	Endonuclease GajA/Old nuclease/RecF-like AAA domain-containing protein	1.0891
8	Peptidoglycan hydrolase	0.8508
9	Ribonucleoside-diphosphate reductase subunit beta	0.824
10	Metallo-beta-lactamase domain protein	0.9978
11	Type VII secretion protein, YukD family	1.11
12	Uncharacterized protein	1.0537
13	Cell wall-binding repeat protein	1.0257
14	Cell wall-binding repeat protein	1.0221
15	Antitoxin	0.5697
16	ABC-2 type transporter transmembrane domain-containing protein	0.1668
17	Ferritin-like protein	1.1171
18	CBS domain protein	0.9741
19	Uncharacterized protein	0.8529
20	Large ribosomal subunit protein uL24	0.6124
21	ACT domain protein	1.0264
22	YjzC-like protein	1.1051
23	NYN domain-containing protein	0.8158
24	CHAP domain protein	0.8252
25	Periplasmic binding protein	0.9848
26	Oxidoreductase, NAD-binding domain protein	1.0851
27	Dipeptidase	0.9934
28	Replication initiation and membrane attachment protein	0.3368
29	Alpha amylase, catalytic domain protein	0.312
30	Polysaccharide biosynthesis protein	0.3913
31	Lipoprotein	0.9981
32	LPXTG-motif cell wall anchor domain protein	1.0466
33	Putative dGTPase	0.9897
34	3D domain protein	1.0095
35	Lipoprotein	0.8274
36	BadF/BadG/BcrA/BcrD ATPase family protein	1.002
37	Ribitol-5-phosphate cytidylyltransferase	0.4136
38	Oxidoreductase, short chain dehydrogenase/reductase family protein	0.3596
39	Zinc-ribbon domain-containing protein	0.721
40	DltD domain protein	0.3036
41	DNA-binding helix-turn-helix protein	0.9929
42	Lipoprotein	0.5198
43	Type VII secretion system accessory factor EsaA	1.0691
44	Uncharacterized protein	1.0868
45	ABC transporter, ATP-binding protein	0.9232
46	Small ribosomal subunit protein uS14	1.0552
47	Large ribosomal subunit protein bL27	1.1145
48	Helix-turn-helix domain-containing protein	0.1066
49	Large ribosomal subunit protein bL20	0.987
50	Flagellar FliJ protein	0.9929
51	Cell wall-binding repeat protein	1.0173
52	Uncharacterized protein	0.0284
53	Response regulator receiver domain protein	1.0023
54	LPXTG-motif cell wall anchor domain protein	1.0137
55	CAAX amino terminal protease family protein	0.5992
56	Small ribosomal subunit protein uS10	0.7333
57	Excalibur domain protein	1.0081
58	Small ribosomal subunit protein bS20	1.0387
59	DivIVA domain protein	1.0225
60	Arylsulfotransferase N-terminal domain-containing protein	0.998
61	Glycosyltransferase, group 2 family protein	1.0253
62	LPXTG-motif cell wall anchor domain protein	1.007
63	Acyl carrier protein	0.0809
64	Uncharacterized protein	0.9996
65	Nucleoside triphosphate/diphosphate phosphatase	0.5432
66	N-acetylmuramoyl-L-alanine amidase	1.0077
67	DNA-directed RNA polymerase subunit epsilon	0.5563
68	YbbR-like protein	1.0947
69	Signal recognition particle protein	0.3172
70	CBS domain protein	0.4193
71	LXG domain-containing protein	0.5803
72	ABC transporter, ATP-binding protein	0.4808
73	Phage major tail protein, phi13 family	0.993
74	RelA/SpoT domain protein	1.0544
75	Transcriptional regulator, GntR family	0.1018
76	Ribosomal protein L7Ae	1.1258
77	4-deoxy-L-threo-5-hexosulose-uronate ketol-isomerase	0.9949
78	DUF2207 domain-containing protein	0.5563
79	LemA family protein	1.0085
80	DUF4176 domain-containing protein	1.1326
81	ESAT-6-like protein	1.0188
82	YqeY-like protein	0.1766
83	Putative hyalurononglucosaminidase	1.0868
84	Small ribosomal subunit protein uS17	0.8802
85	Outer surface protein	0.9101
86	Pilin isopeptide linkage domain protein	1.0169
87	Bleomycin resistance protein	1.011
88	Sortase family protein	1.0526
89	Transposase	1.0859
90	Cell envelope-like function transcriptional attenuator common domain protein	1.0305
91	GtrA-like protein domain-containing protein	0.9072
92	Lipoprotein	0.9882
93	Large ribosomal subunit protein bL33	0.9998
94	DNA-directed RNA polymerase subunit omega	1.1517
95	Ribonuclease P protein component	0.3267
96	Acyl carrier protein	0.3664
97	DNA-binding helix-turn-helix protein	1.0418
98	Phosphopantetheine adenylyltransferase	1.0486
99	Trypsin	0.9085
100	DnaD domain protein	0.9444
101	Putative ACR, COG1399	0.4689
102	Pyridinium-3,5-bisthiocarboxylic acid mononucleotide nickel insertion protein	0.8938
103	Response regulator receiver domain protein	0.6493
104	PTS system mannose/fructose/sorbose family IID component	0.7343
105	DUF1310 family protein	1.0365
106	Oxidoreductase, short chain dehydrogenase/reductase family protein	0.5302
107	Citrate lyase acyl carrier protein	0.826
108	Small ribosomal subunit protein bS18	1.0776
109	Small ribosomal subunit protein uS15	0.6017
110	LPXTG-motif cell wall anchor domain protein	1.0351
111	Putative cross-wall-targeting lipoprotein signal	1.0018
112	Uncharacterized protein	0.9772
113	Putative DNA/RNA non-specific endonuclease	0.9956
114	Transcriptional regulator, Crp/Fnr family	0.4498

## Data Availability

Raw mass spectrometry data have been deposited in the PRIDE repository (Project ID: PXD059541; username: reviewer_pxd059541@ebi.ac.uk; password: gyAZOgM9BoHt).

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
