# Peer review of "Differential Proteomic Analysis of Extracellular Vesicles Produced by *Granulicatella adiacens* in Biofilm vs. Planktonic Lifestyle"

_dentistry, 2025, doi:10.3390/dj13120557_

Round 1

Reviewer 1 Report

Comments and Suggestions for Authors

The manuscript addresses a topic of great interest: the proteomic characterization of extracellular vesicles (EVs) of Granulicatella adiacens in biofilm and planktonic conditions. The study provides new and potentially relevant data on the role of EVs as virulence mechanisms. The work is well structured, with a detailed methodology and results supported by robust bioinformatic analyses (PCA, GSEA, hierarchical clustering, etc.). The discussion section appropriately interprets the data and links their clinical relevance.
However, there are some critical points that deserve attention before potential publication.
Strengths
Originality: first comparative proteomic analysis of G. adiacens EVs in biofilm and planktonic conditions.
Robust methodology: use of high-resolution LC-MS/MS, advanced bioinformatics, well-performed biological controls.
Clinical interpretation: connection with infective endocarditis and the role of EVs in immune modulation.
Extensive data: identification of over 1,000 proteins and in-depth functional analyses (virulence, GO terms, PPI networks).
Major critical issues
First and foremost: Clarity of objectives
The introduction is very extensive, but the research question is not formulated concisely.
It would be useful to better explain the central hypothesis: whether biofilm-derived EVs differ functionally from planktonic EVs and how this contributes to virulence.
Add all natural and technological substances that can modify the microbiota, such as laser therapy in periodontal patients (Pardo et al. Applied Sciences).

Only EVs from a reference strain (G. adiacens CCUG 27809) are used. The use of at least a second clinical strain would increase the robustness of the results.
Functional controls on cells unstimulated with heat-treated or protease-digested EVs are lacking to demonstrate that cytokine induction is truly protein-dependent.
Proteomic Analysis
The significance criteria (p < 0.05 and log2FC > 1.5) are correct, but a complete table of the main top hits with FDR and fold-change values ​​is not shown.
Some GSEA pathways have very low enrichment scores: interpretation should be more cautious.
The discussion of the 22 (biofilm-specific) and 17 (planktonic-specific) proteins should be expanded, explaining their virulence relevance.
Immunological Section
Only three cytokines (IL-8, IL-1β, IL-17) are analyzed. Given their immune relevance, it would be desirable to include TNF-α, IL-6, or chemokines.
The findings on IL-17 are quickly dismissed: it should be explained why this cytokine, crucial for mucosal processes, is not modulated.
Clinical Connection
Although the association with endocarditis is mentioned, there is no discussion of how EV protein profiles may specifically contribute to endocardial colonization or treatment resistance. What about the role in neutrophils? (Bassani et al, ijms)
Minor Issues
Some typos and inconsistencies: "Lifestyle" in the title, "IL-1" in the text.
The figures are well done but very dense: some could be moved to a supplement, leaving only the key graphs in the text.
The methodological description is very detailed; however, some information (type of centrifuge, spectrometer model) could be abbreviated.
References: The most recent citations (2022–2024) on the role of EVs in Gram-positive bacteria are not included.

IN SUMMARY
Strengthen the introduction with a clear working hypothesis.
Add or better discuss methodological limitations (single strain, limited set of cytokines).
Expand the discussion of specific data (unique proteins, virulence factors).
Improve the connection to clinical and therapeutic implications.
Revise language, typos, and figure layout.
Conclusion
The manuscript is of good quality and presents interesting data relevant to oral microbiology and infectious disease. With some substantial revisions and improved clarity, it could be suitable for publication.

Author Response

REVIEWER 1

The manuscript addresses a topic of great interest: the proteomic characterization of extracellular vesicles (EVs) of Granulicatella adiacens in biofilm and planktonic conditions. The study provides new and potentially relevant data on the role of EVs as virulence mechanisms. The work is well structured, with a detailed methodology and results supported by robust bioinformatic analyses (PCA, GSEA, hierarchical clustering, etc.). The discussion section appropriately interprets the data and links their clinical relevance.

However, there are some critical points that deserve attention before potential publication.

Strengths

Originality: first comparative proteomic analysis of G. adiacens EVs in biofilm and planktonic conditions.

Robust methodology: use of high-resolution LC-MS/MS, advanced bioinformatics, well-performed biological controls.

Clinical interpretation: connection with infective endocarditis and the role of EVs in immune modulation.

Extensive data: identification of over 1,000 proteins and in-depth functional analyses (virulence, GO terms, PPI networks).

Major critical issues

First and foremost: Clarity of objectives

The introduction is very extensive, but the research question is not formulated concisely.

It would be useful to better explain the central hypothesis: whether biofilm-derived EVs differ functionally from planktonic EVs and how this contributes to virulence.

Response: We thank the reviewer for the suggestion. The introduction has been thoroughly revised. Hypothesis of the study is now clarified, and objectives are clearly stated.

Add all natural and technological substances that can modify the microbiota, such as laser therapy in periodontal patients (Pardo et al. Applied Sciences).

Response: The following text concerning substances/factors that modify oral microbiota is now added in the introduction.

“Furthermore, external interventions such as antimicrobial agents (Dagli, Dagli et al. 2016, Brookes, Belfield et al. 2021, Kleine Bardenhorst, Hagenfeld et al. 2024) and photobiomodulation (e.g., low-level laser therapy) (Alhazmi 2023), can alter oral microbiota composition and host responses, potentially influencing EV production and signaling, underscoring their clinical significance.”

Only EVs from a reference strain (G. adiacens CCUG 27809) are used. The use of at least a second clinical strain would increase the robustness of the results.

Response: We agree with the reviewer that inclusion of a second clinical strain would have increased the robustness of the results. We have addressed this issue as a limitation of this study. Our future investigations, utilizing multiple clinical strains, may address the issue of intraspecies (strain-level) variability.

Functional controls on cells unstimulated with heat-treated or protease-digested EVs are lacking to demonstrate that cytokine induction is truly protein-dependent.

Response: We understand the reviewer’s concern. The EVs are packed not only with proteins but also lipids and lipoproteins. We have addressed this in the limitations of the study as below:

“The other limitation is that functional control experiments using heat- or protease-treated EVs were not performed in this study and will be addressed in future work to confirm that cytokine induction is protein-mediated”

Proteomic Analysis

The significance criteria (p < 0.05 and log2FC > 1.5) are correct, but a complete table of the main top hits with FDR and fold-change values ​​is not shown.

Response: A complete table (Table S2) consisting of main top hits with FDR and fold-change values is provided as a supplementary file in our revised submission.

Some GSEA pathways have very low enrichment scores: interpretation should be more cautious.

Response: Thank you for the suggestion. We have exercised caution accordingly.

The discussion of the 22 (biofilm-specific) and 17 (planktonic-specific) proteins should be expanded, explaining their virulence relevance.

Response: Virulence relevance of the unique proteins has been elaborated now. The following text is added in the revised manuscript:

“This near-complete overlap implies that the shift between these lifestyles relies primarily on quantitative changes (up- or down-regulation) in a shared proteome rather than a largescale protein replacement. However, unique proteins in each state, though small in number—could play pivotal roles in biofilm formation or planktonic lifestyle, acting as specialized molecular switches for these distinct modes of existence. Biofilm-specific proteins such as magnesium transporter (mgtE) and Cyclic-di-AMP phosphodiesterase have been shown to play important roles in the regulation of type III secretion system in different bacteria (Chakravarty, Melton et al. 2017, Gutierrez, Wong et al. 2022). Further, deletion of mutL, a protein involved in DNA mutation repair was found to affect adhesion and biofilm formation leading to an overall reduced virulence (Placzkiewicz, Adamczyk-Poplawska et al. 2019). Proteins with metallo-beta lactamase domain can confer resistance to a broad range of beta-lactam antibiotics. Its presence in biofilm vesicles indicates a mechanism for antibiotic inactivation(Diene, Pontarotti et al. 2023).”

Immunological Section

Only three cytokines (IL-8, IL-1β, IL-17) are analyzed. Given their immune relevance, it would be desirable to include TNF-α, IL-6, or chemokines.

The findings on IL-17 are quickly dismissed: it should be explained why this cytokine, crucial for mucosal processes, is not modulated.

Response: We agree with the reviewer’s concern. We will certainly include TNF-alpha, IL-6 etc. in our future studies. This has been added as a limitation of the study in the discussion section. Significance of IL-17 has also been thoroughly discussed now. The following text has been added:

“Interestingly, IL-17 levels remained unchanged, suggesting a targeted immune modulation rather than broad activation. IL-17 is an important cytokine in the mucosal immunity. The observed similarity in IL-17 stimulation by biofilm and planktonic EVs is likely because they share similar outer envelope components, such as bacterial cell wall elements and surface proteins, which are known to induce this key mucosal immunity cytokine (Weber, Zimmermann et al. 2014, Turner, Raisley et al. 2023).The observed cytokine patterns are relevant to G. adiacens pathogenesis, given the roles of IL-8 and IL-1β in infective endocarditis and oral infections (Yamaji, Kubota et al. 1995, Araujo, Ferrari et al. 2015).While TNF-α and IL-6 were not assessed in this study, future cytokine panels may include these key inflammatory mediators to better define the immunomodulatory spectrum of G. adiacens EVs.”

Clinical Connection

Although the association with endocarditis is mentioned, there is no discussion of how EV protein profiles may specifically contribute to endocardial colonization or treatment resistance. What about the role in neutrophils? (Bassani et al, ijms)

Response: We have elaborated the clinical relevance of EVs in the discussion part now. The following text has been added in the discussion:

“In contrast to their soluble counterparts, vesicle-associated virulence factors are uniquely shielded from degradation by host proteases (Kuehn and Kesty 2005). Furthermore, vesicles facilitate the targeted delivery of these factors as concentrated packages to host cells and tissues, thereby amplifying damage at specific sites. Similar to oral bacteria implicated in infective endocarditis—such as Aggregatibacter actinomycetemcomitans (Thay, Damm et al. 2014), Kingella kingae (Maldonado, Wei et al. 2011), and others (Yumoto, Hirota et al. 2019, Alkandari, Bhardwaj et al. 2020, Iwabuchi, Yoshida et al. 2025)—it is plausible that G. adiacens similarly utilizes its extracellular vesicles, which are enriched with numerous putative virulence proteins, in the pathogenesis of this infection.”

Minor Issues

Some typos and inconsistencies: "Lifestyle" in the title, "IL-1" in the text.

Response: Thanks so much. Typo inconsistencies have been corrected.

The figures are well done but very dense: some could be moved to a supplement, leaving only the key graphs in the text.

Response: Figures have been reorganized now, moving some of the panels in the figures to the supplementary figures file.

The methodological description is very detailed; however, some information (type of centrifuge, spectrometer model) could be abbreviated.

Response: Thank you. Details such as centrifuge names and spectrophotometer models have been shortened.

References: The most recent citations (2022–2024) on the role of EVs in Gram-positive bacteria are not included.

Response: Several references from EV literature during 2022-2024 are now cited in the text.

IN SUMMARY

Strengthen the introduction with a clear working hypothesis.

Add or better discuss methodological limitations (single strain, limited set of cytokines).

Expand the discussion of specific data (unique proteins, virulence factors).

Improve the connection to clinical and therapeutic implications.

Revise language, typos, and figure layout.

Response: We sincerely thank the reviewer for his/her valuable time and efforts in critically assessing this manuscript. All concerns raised have been addressed and the manuscript revised accordingly.

Conclusion

The manuscript is of good quality and presents interesting data relevant to oral microbiology and infectious disease. With some substantial revisions and improved clarity, it could be suitable for publication.

Reviewer 2 Report

Comments and Suggestions for Authors

The article presents a comparative proteomic analysis of extracellular vesicles. The study is very good and provides insights into how bacterial EVs contribute to adaptation, immune modulation. However, the article needs some improvement.

  1. The introduction can be improved by comparing EV proteomics in biofilm vs. planktonic states should be emphasized more clearly.
  2. The study only uses 3 biological replicates per condition which is acceptable for proteomics but this limitations should be acknowledged in the discussion.
  3. SEM imaging and SDS page are well presented but more detailed EV characterization can be discussed in the limitations
  4. The cytokine assays using the PBMCs showed no change in IL-17. Authors needs to discuss why IL-17 was included and whether other cytokines such as TNF-alpha or IL-6 was measured or not?
  5. Please add some future directions.
  6. Figures are clear but some heatmaps and volcano plots are difficult to read so, please enlarge the font size to make it reader friendly.
  7. The venn diagram make it simple only focusing on unique proteins.
  8. The supplementary file is difficult to read so please provide brief guide on it. 

Author Response

REVIEWER 2

The article presents a comparative proteomic analysis of extracellular vesicles. The study is very good and provides insights into how bacterial EVs contribute to adaptation, immune modulation. However, the article needs some improvement.

The introduction can be improved by comparing EV proteomics in biofilm vs. planktonic states should be emphasized more clearly.

Response: The introduction has been modified, by adding more literature on EV proteomics in biofilm vs planktonic states. The following text is added now in the introduction:

“Similar to proteomic differences between biofilm and planktonic cells (De Angelis, Siragusa et al. 2015, Dumitrache, Klingeman et al. 2017), proteomic analyses comparing EVs from biofilm and planktonic conditions have highlighted quantitative differences in composition that likely reflect the physiological conditions (Marinacci, D'Ambrosio et al. 2025). In the context of oral cavity, previous studies have profiled the secretome and EV proteome of G. adiacens (Karched, Bhardwaj et al. 2019, Alkandari, Bhardwaj et al. 2020), yet the impact of bacterial lifestyle—biofilm versus planktonic growth—on EV composition and virulence potential remains unexplored.”

The study only uses 3 biological replicates per condition which is acceptable for proteomics but this limitations should be acknowledged in the discussion.

Response: This issues has been addressed in the “limitations and future directions” part.

SEM imaging and SDS page are well presented but more detailed EV characterization can be discussed in the limitations

Response: This shortcoming has been discussed in the “limitations and future directions” section. 

The cytokine assays using the PBMCs showed no change in IL-17. Authors needs to discuss why IL-17 was included and whether other cytokines such as TNF-alpha or IL-6 was measured or not?

Response: This has been elaborated in the discussion as well as in the limitations section. The following text is now added in the discussion:

“The observed similarity in IL-17 stimulation by biofilm and planktonic EVs is likely because they share similar outer envelope components, such as bacterial cell wall elements and surface proteins, which are known to induce this key mucosal immunity cytokine (Weber, Zimmermann et al. 2014, Turner, Raisley et al. 2023).The observed cytokine patterns are relevant to G. adiacens pathogenesis, given the roles of IL-8 and IL-1β in infective endocarditis and oral infections (Yamaji, Kubota et al. 1995, Araujo, Ferrari et al. 2015).While TNF-α and IL-6 were not assessed in this study, future cytokine panels may include these key inflammatory mediators to better define the immunomodulatory spectrum of G. adiacens EVs.”

Please add some future directions.

Response: Thank you. Future directions are added in the discussion section.

Figures are clear but some heatmaps and volcano plots are difficult to read so, please enlarge the font size to make it reader friendly.

The venn diagram make it simple only focusing on unique proteins.

Response: Figures have been thoroughly reorganized, moving some of them to the Suplementary Figures file.

The supplementary file is difficult to read so please provide brief guide on it.

Response: A brief guide has been added in the adjacent worksheet named “Guide” in the same excel file.

Reviewer 3 Report

Comments and Suggestions for Authors
  1. To add a link: although many EV studies have focused on Gram-negative, there is a growing consensus about the existence of EV in Gram-positive, which motivates the study of EV specifically in G. adiacens.
  2. Please pay attention to the terminology. Work on the uniformity of terminology. For example, “vesicle preparations” vs “EV preparations”
  3. n conclusion, the promising areas of further research should be outlined in more detail. It is necessary to emphasize the clinical and/or biomedical significance of the study. How can the results obtained contribute to the development of new therapeutic approaches and diagnostic methods?

Author Response

REVIEWER 3

  1. The introduction must explicitly articulate the objectives and novel contributions of the study.

Response: The introduction has been thoroughly revised, clealrly stating the objectives and contributions of the study.

  1. What scientific hypothesis do you intend to verify or investigate?

Response: We hypothesised that EVs released by G. adiacens during biofilm growth differ proteomically and functionally from those produced in planktonic conditions, contributing to distinct mechanisms of virulence and host immune modulation. This has now been added in the introduction.

  1. The introduction must adhere to a structured approach, including a justification for the relevance of the topic, an identification of gaps in existing knowledge, and a clear formulation of the research objectives.

Response: Thank you for the critical comment. We have thoroughly revised the introduction now, addressing gaps in knowledge and clearly stating the objectives.

  1. An analysis of the methodology employed in the study revealed that it was only partially disclosed. Specific parameters utilized for comparison were not specified, nor were the criteria employed for evaluation provided.

Response: Details of the specific parameters used in generating the figures were different and therefore, are provided in figure legends for each figure.

  1. The Results section does not furnish sufficient statistical evidence to either support or refute the hypothesis.

Response: Hypotheses of the study has been clarified now. Statistical values presented in the results text as well as figure legends.

  1. The findings of the study exhibit a lack of clarity and precision in the formulation of conclusions, which may impede their interpretation and subsequent application for scientific and practical purposes.

Response: We apologize for being unclear. The text concerning the study conclusions has been modified now.

To add a link: although many EV studies have focused on Gram-negative, there is a growing consensus about the existence of EV in Gram-positive, which motivates the study of EV specifically in G. adiacens.

Response: Thank you. We agree with the reviewer. Accordingly, more literature on EVs has been added now.

Please pay attention to the terminology. Work on the uniformity of terminology. For example, “vesicle preparations” vs “EV preparations”

Response: The text has now been revised to keep it consistently “EV preparation”.

n conclusion, the promising areas of further research should be outlined in more detail. It is necessary to emphasize the clinical and/or biomedical significance of the study. How can the results obtained contribute to the development of new therapeutic approaches and diagnostic methods?

Response: Conclusions are revised now to include clinical significance, contribution to the development of new therapeutic approaches and diagnostic methods.

Round 2

Reviewer 1 Report

Comments and Suggestions for Authors

I thank the authors for reviewing the manuscript for almost all of its comments. Please re-format the article in the journal format and add the suggested references to improve the quality of the manuscript.